# Boosting Sustainable Operations with Sustainable Supply Chain Modeling: A Case of Organizational Culture and Normative Commitment

**DOI:** 10.3390/ijerph191711131

**Published:** 2022-09-05

**Authors:** Sebastjan Lazar, Vojko Potočan, Dorota Klimecka-Tatar, Matevz Obrecht

**Affiliations:** 1Faculty of Logistics, University of Maribor, Mariborska Cesta 7, 3000 Celje, Slovenia; 2Faculty of Economics and Business, University of Maribor, Razlagova 14, 2000 Maribor, Slovenia; 3Faculty of Management, Czestochowa University of Technology, Al. Armii Krajowej 19b, 42-200 Czestochowa, Poland

**Keywords:** logistics, supply chain, sustainable development (SD), sustainable development dimensions, sustainable development goals (SDGs), organizational culture (OC), normative commitment (NC)

## Abstract

The importance of sustainability in supply chain management is growing worldwide. It is possible to find reasons for this using various phenomena that negatively affect humanity, e.g., climate change, scarce materials, supply disruptions, and complex fossil fuel dependency. Because of that, is extremely important to constantly look for new ways to systematically increase sustainability in enterprises and their logistics and supply chain processes by considering different stakeholders and influential factors. Therefore, this paper explores how different types of organizational culture and normative commitment impact sustainability and each other in business logistics and supply chains and develops a conceptual model to manage this challenge. Gaining new insights is valuable especially for managers to obtain better information on how to improve sustainability not just by integrating green technologies but mainly by changing culture, attitude, and perception in their enterprises. The research is focused on employees from global logistics or related branches in micro, small, medium, and large enterprises with the primary activity mostly related to manufacturing, transport, and storage. The findings are based on the questionnaire which was sent directly to 1576 employees from 528 enterprises. A total of 516 employees from enterprises that are mostly located in 34 countries responded to requests for participation. The results reveal statistically significant positive and negative impacts, e.g., clan culture has a positive statistically significant impact on the sustainable development of supply chains. Most of the connections to the eighth Sustainable Development Goal by the United Nations (decent work and economic growth) were also found, which was the enterprise’s highest priority with a share of 52.99%. A contribution to the theory development is gained using the developed model that considers both positive and negative statistically significant impacts studied.

## 1. Introduction

### 1.1. Background

Human civilization has been growing since its inception, causing severe disturbances to the earth’s system and may even lead to future global collapse [1]. There are several interrelated factors related to the growth of the world’s population, e.g., greater resource consumption and greater waste generation [2]. This also makes it essential to protect the environment around the world in order to ensure social and economic development for the benefit of present and future generations [3]. Sustainable development (SD) is considered an important concept in the context of the common future of humanity [4] and has, at the same time, a huge impact on enterprises and their supply chains [5]. Sustainability and SD are also known as one of the primary perspectives that can promote well-being in enterprises [6]. As a result, SD is now considered a key concept and solution in creating the future of humanity [7] and has gained worldwide attention in recent years [8]. Its recognition is seen also in the 17 United Nations Sustainable Development Goals—SDGs [9]. Sustainable challenges constantly place pressure on states, the market, and civil society to take action [10]. Enterprises must continually adjust their operations [11] because they are facing increasingly intense environmental pressure and market competition [12]. Enterprises are also changing business models in order to adapt to the rapidly changing environment [13,14] and because of this it is important to constantly develop new models that improve sustainability in enterprises and their supply chains, including the logistics sector. It was also pointed out [15] that normative commitment (NC) improves all types of logistics and consequently supply chains, while it was also noted [16] that the role of organizational culture (OC) supports the vision of sustainability. As a result, this research is focused on the concept of improving sustainability in logistics and supply chains, considering the impact of OC and NC.

### 1.2. Literature Review

The logistics sector is developing extremely fast [17] and its development is related to the development of industry [18]. Logistics was also highlighted as an important element in supply chains [19,20] combining logistics operations of different enterprises. Supply chains are becoming increasingly globalized, complex [21], and subject to uncertainty [22]. Their aim is to improve customer satisfaction and overall competitiveness in the global marketplace [23]. On a global level, supply chains must quickly adapt to the changing world because of new situations such as the COVID-19 pandemic [24], the war in Ukraine [25], or climate change [26]. Convergence between the supply chain and sustainable operation was also defined [27], which was first focused narrowly (locally) and then transferred to the entire supply chain.

Sustainable operation and consequently SD has become quite popular today in modern development discourse [28] and is an inevitable trend in many areas [29], including supply chains [30]. Its role is growing rapidly [31], which is also confirmed by many studies, e.g., [32,33]. SD was defined [34] as a way of meeting the needs of the current population without jeopardizing the ability of successors to meet their needs. There are three dimensions of SD in the world, namely: social development, economic development, and environmental development [35], which are also called the main variables of SD [36]. The United Nations has set an agenda for SD until 2030, which includes 17 SDGs [37,38], which are also associated with all three dimensions of SD [39]. These goals immediately became an unprecedented global compass to cope with current sustainable challenges [40]. Their aim is to achieve a better and more sustainable future at the global level [41] and at the same time cover all sectors [42], including global supply. Their importance is proven by the fact that in the year 2015, 193 countries committed to SDGs implementation [43]. Because of that, it has great historic significance [44]. These goals are also a strategy to promote sustainable practices and solutions that addresses the main issues facing our entire society [45], also logistics and non-logistics enterprises and their supply chains. Focusing on sustainability in supply chains is a big step towards the wider acceptance of SD as a new development paradigm [46]. Sustainable operation in supply chains has become extremely important [47] because it also improves the image of enterprises on the social front [48] and even improves business benefits [49]. It was also mentioned [50] that the path to the adoption of sustainability principles in enterprises (and consequently in their supply chains) leads to the adoption of a sustainable OC.

An important construct in times of change is also OC. It is also an important part of sustainability [51], especially because every enterprise has it [52]. The importance of OC for ensuring the sustainability of the enterprise was also emphasized [53]. OC is also important for strengthening green practices in the workplace [54]. OC is extremely dynamic and complex [55] and consequently receives attention in all areas [56], including, e.g., task performance [57]. OC was also defined as a way that we do a certain thing here or in a certain enterprise [58,59]. A particular emphasis is placed on its effect on employee behavior [60] and its profound impact on various situations [61,62]. OC can be measured and there are differences in OC and different types of OC between business entities [63]. The great importance of knowing OC in order to make important decisions and subsequently introduce various improvements is also highlighted [64].

There are different levels of individual commitment, including NC, across cultures [65]. Commitment is one of the most important elements in enterprises [66], which is also key to success in today’s business world [67]. An important link between sustainability and commitment in enterprises was also highlighted [68]. Organizational commitment is defined as the relationship between employees and their enterprises [69]. It was also exposed that commitment to the enterprise develops from job satisfaction [70]. Organizational commitment is a key factor in creating a high level of performance and a low level of absence and fluctuation in the business environment [71] and is also defined as a psychological state, which comprises three distinct components for retaining employment and manifests itself as: the desire for affective commitment, the need for continuance commitment, and the obligation for NC [72]. The literature particularly highlights the importance of loyalty to the enterprise and highlights the personality of the individual and the possible influence of the legal entity on the commitment [73]. Employees with a high level of NC believe that they should stay in the organization [74]. It was found that it has a direct and indirect impact on employee motivation in enterprises [75]. Employees with a higher level of commitment are more motivated and consequently spend more energy for the enterprise in which they are employed [76]. The importance and impact of commitment in supply chains in modern situations that we face, e.g., the COVID-19 pandemic, was also particularly highlighted [77].

### 1.3. Research Gap, Question, and Goals

It was mentioned that more and more enterprises are striving for sustainable management in supply chains to improve business efficiency [78]. It is also exposed [79] that enterprises are taking sustainable measures also due to policy pressures and environmental pacts. Enterprises and their managers are increasingly thinking about ways to make their business models as sustainable as possible [80]. Enterprises can use different models to achieve SD [81]. The model that examines the dependencies of how aspects of corporate social responsibility affect SD was exposed [82]. The ISO 14,001 was also mentioned [83]. OC and leadership in the sustainability of small businesses were analyzed [84]. The impact of OC on sustainable performance in the hotel sector was highlighted [85]. The positive connection between higher commitment and employee perceptions of environmental management practices in food-processing businesses was also pointed out [86].

With respect to the existing literature, a lack of research in the field of the impact of OC and NC on sustainable supply chains was identified. At the same time, we observed a large shortage of models for improving sustainability in the supply chain through OC and NC. We also noticed a deficit of studies in the field of linking individual SDGs with logistics and supply chains and in identifying the most common in micro, small, medium, and large enterprises. Consequently, this study focuses on the analysis of the impacts of OC and NC on supply chain sustainability and research on the current status of 17 SDGs in terms of priorities in micro, small, medium, and large logistics and non-logistics enterprises. A new model is being developed to propose the potential application of results in practice with the goal to contribute to a brighter future in business logistics.

According to the presented gap, our research questions are the following: (1) To which dimensions and goals of SD are logistics and related enterprises most focused? (2) What is and how do OC and NC affect the SD of supply chains? (3) How does OC affect NC? (4) How can the impact of OC and NC on supply chain sustainability be integrated into development of a new model for integration into practice?

The aim of the research is to examine the connections in terms of the priorities of individual SDGs and to identify the most common in micro, small, medium, and large logistics and non-logistics enterprises. At the same time, the aim is to develop a model for improving sustainable supply chains from a logistics perspective, considering the impact of OC and NC and their inter-relations.

The study is structured as follows: (a) materials and methods; (b) results section, divided into: SD (dimensions and goals); OC and NC—also the impact of individual areas with the development of the model and related potential results application; (c) discussion and interpretation; and (d) concluding remarks.

## 2. Materials and Methods

### 2.1. Instrument Used

For the purpose of the research, we prepared a questionnaire with the help of various sources, which contained the following sets: (a) demographic characteristics, modified from [87,88,89,90,91]; (b) SD, modified from [35,41,92,93,94,95,96]; (c) OC, modified from [97,98], considering the OC Assessment Instrument which is a well-known research method worldwide used to examine OC and has four types of cultures—clan (collaborative orientation), adhocracy (creative orientation), market (competing orientation), and hierarchy (controlling orientation) [97]; and (d) NC, modified from [99,100], considering both reflections of NC behavior that exist: direct and indirect [100]. Before starting research data collection, the questionnaire was pilot tested to avoid possible ambiguities, therefore, an experimental method was used in order to test the understanding of the questionnaire [101]. The responses were recorded mainly using the five-point Likert scale. Consent rates ranged from one, which meant disagreement at all, to a maximum of five, which meant very strong agreement. A Likert five-point scale was also used in other related surveys to measure attitudes, e.g., [102].

### 2.2. Data Collection, Sample, and Procedures

The data for analysis was obtained from the questionnaire. The questionnaire was conducted using the 1ka online tool for the survey. The target group of questionnaires was employees who have been involved in logistics for at least 3 years and were employed in logistics and non-logistics micro, small, medium, and large enterprises. The questionnaire was sent directly to 1576 employees from 528 enterprises. The survey was conducted in the last quarter of the year 2021. During the survey period, we obtained 516 surveys. Demographic characteristics are shown in Table 1.

### 2.3. Data Analysis and Procedures

For the purpose of the research, we set the following hypotheses and sub-hypotheses:

**H1.** 
*OC has a statistically significant impact on the SD of the supply chain from the logistics perspective.*


**H1a.** 
*Clan culture has a statistically significant impact on the SD of the supply chain from the logistics perspective.*


**H1b.** 
*Adhocracy culture has a statistically significant impact on the SD of the supply chain from the logistics perspective.*


**H1c.** 
*Market culture has a statistically significant impact on the SD of the supply chain from the logistics perspective.*


**H1d.** 
*Hierarchy culture has a statistically significant impact on the SD of the supply chain from the logistics perspective.*


**H2.** 
*NC of employees has a statistically significant impact on the SD of the supply chain from the logistics perspective.*


**H2a.** 
*Indirect NC has a statistically significant impact on the SD of the supply chain from the logistics perspective.*


**H2b.** 
*Direct NC has a statistically significant impact on the SD of the supply chain from the logistics perspective.*


**H3.** 
*OC has a statistically significant impact on NC.*


**H3a.** 
*Clan culture has a statistically significant impact on indirect and direct NC.*


**H3b.** 
*Adhocracy culture has a statistically significant impact on indirect and direct NC.*


**H3c.** 
*Market culture has a statistically significant impact on indirect and direct NC.*


**H3d.** 
*Hierarchy culture has a statistically significant impact on indirect and direct NC.*


Obtained quantitative data were processed with specialized tools for statistical processing, e.g., Statistical Package for the Social Sciences.

For the purpose of testing hypotheses H1, H2, H3, and their sub-hypotheses, we first reduced the number of variables/items/statements with factor analysis (principal components method) to a manageable number of factors with which we can perform statistical processing, for which we took the methodology from [103,104]. Before performing the factor analysis, we checked the adequacy of the connectivity between the variables for inclusion/use in the factor analysis with the Pearson correlation coefficient. In the process of factor analysis, the validity of the measured factor was checked by using the Bartlett sphericity test (verification of the adequacy of variables for further factor analysis), KMO (Kaiser Meyer Olkinova) the degree of sample adequacy, utility, and share of the variability of the dependent variable explained by the independent variable. For the formed factors, reliability was examined with Cronbach’s alpha coefficient. For the examination of each formed factor reliability, the following criteria of alpha value were used: (1) to 0.50 = unacceptable, (2) from 0.50 to 0.60 = poor, (3) from 0.60 to 0, 70 = questionable, (4) from 0.70 to 0.80 = acceptable, (5) from 0.80 to 0.90 = good, (6) above 0.90 = excellent. Cronbach’s alpha coefficient results were the following for formed and used factors: SD (0.86), OC (0.83), clan culture (0.89), adhocracy culture (0.82), market culture (0.79), and hierarchy culture (0.73).

Hypotheses H1, H2, and H3 (and all their sub-hypotheses) were tested using the statistical method of linear regression analysis if their direct impact was statistically significant. If we talk about the ratio of one variable, such a procedure is called simple regression analysis, if we talk about the ratio for more than one variable, such a procedure is called multiple regression analysis [105]. In the process of performing a linear regression analysis, the quality of the formed regression model was checked (with ANOVA), the share of the explained variability of the dependent variable (with adjusted R^2^), and then the statistical significance, strength, and direction of impacts (standardized Beta coefficient, t statistics, and its value *p*). In hypothesis H1 and all its sub-hypotheses, the data were analyzed using simple regression analysis (one independent variable and one dependent variable). All variables/items/statements that were included in the linear regression analysis under hypothesis H1 and sub-hypotheses H1a, H1b, H1c, and H1d were factors, therefore, new variables/items/statements that were previously formulated by factor analysis (principal components method). In hypothesis H2 and both of its sub-hypotheses, the data were analyzed using multiple regression analysis (several independent variables and one dependent variable). When testing hypothesis H2 and both of its sub-hypotheses, we used a pre-formed factor with factor analysis (principal components method) for SD, while these could not be designed with sufficient quality for the area of NC. Consequently, the area of NC was analyzed on the basis of individual statements/variables. In hypothesis H3 and all its sub-hypotheses, the data were analyzed using simple regression analysis (one independent variable and one dependent variable). When testing hypothesis H3, we used pre-formed factors with factor analysis (principal components method) for OC and its types, while these could not be designed with sufficient quality for the area of NC. Consequently, the area of NC was analyzed on the basis of individual statements/variables. Since the area of NC in the regression analysis was a dependent variable, we designed 8 regression models for hypothesis H3 and all its sub-hypotheses.

Based on research, data analysis, and synthesis of findings, we developed a model for the development of a sustainable supply chain from a logistics perspective, which is considering the impact of OC and NC. After detailed data processing, the results were analyzed and interpreted in the following structure: (a) SD (dimensions and goals); (b) OC; (c) NC; (d) the impact of individual areas and development of the model; and (e) potential results application from the new model.

## 3. Results

### 3.1. Sustainable Development

Figure 1 shows the average values of the dimensions of SD in sample enterprises. Among age groups, the social dimension was equally distributed, but the economic dimension was best assessed by respondents aged 51 and over, while the dimension of environmental development was best assessed by people aged 41 or over. People with a higher education rated the dimension of social development the best, while people with basic/lower education surprisingly gave the lowest score in the dimension of economic development. In the dimension of environmental development, the highest marks were given by persons who do not have a high school, diploma, Master’s degree, MBA, or Ph.D. The persons employed in the top management gave the best marks in all three dimensions in comparison with the others (everywhere they are followed by the persons in the middle management in the second place according to the best marks) which shows awareness and prioritizing SD on the executive level. The highest assessment of the dimension of social and environmental development was given from persons with a total length of employment of up to 3 years. The economic dimension was most assessed by persons with up to 3 years of total employment and persons with total employment of 20 years or more (both with identical average values). Employees with a length of service from 10 to 20 years in the current enterprise rated the social development dimension the best. Respondents from small enterprises rated the dimension of social development the best, while the dimension of economic development was rated equally and the highest in small, medium, and large enterprises. In the dimension of environmental development, the highest average value was given by persons from micro and large enterprises. Respondents from enterprises where the primary activity is transport and storage (e.g., transport enterprise) rated the dimension of social development the best. Persons employed in enterprises with a primary manufacturing activity (e.g., manufacturing enterprise) rated the dimension of economic development and the environmental dimension as the highest. Respondents who do not think about changing employers rated the social dimension the best.

Figure 2 shows the overall results of connections to individual SDGs in sample enterprises, where it was found that most of the connections were to the eighth SDG by the United Nations (decent work and economic growth).

### 3.2. Organizational Culture

Figure 3 shows the current status of OC considering its six key dimensions in sample enterprises for the overall results.

Figure 4 shows the current status of OC by its types in sample enterprises.

For people under the age of 30 and for people aged from 41 to 50, the clan culture is at the forefront. From the ages of 31 to 40, the leading role with identical average values has the following cultures: clan, market, and hierarchy. Clan and hierarchy cultures, on the other hand, play a leading role for people aged 51 and over. Clan culture and hierarchy culture play a leading role among people with a Master’s degree, MBA, or Ph.D., and among people who completed high school. Clan culture also plays a leading role among people with a degree and people with other education. Clan culture is also the most characteristic of top and middle management. In other positions, the following types of culture have a leading role with an identical average value: clan culture, market culture, and hierarchy culture. In terms of the total length of service, the culture of the clan is always at the forefront (for employees with a total length of service from 3 to 10 years, the first place is shared with the culture of hierarchy). Even in the total length of service in the current enterprise, the culture of the clan is always at the forefront (for employees with a total length of service in the current enterprise from 3 to 10 years and more than 20 years, clan culture shares first place with the culture of hierarchy). Clan culture is most characteristic for micro and small enterprises, while hierarchy culture is most typical for medium and large enterprises. Manufacturing activities (e.g., manufacturing enterprise) have the same average value in a leading position for clan, market, and hierarchy culture. In other activities, the culture of the clan is at the forefront. The culture of hierarchy is leading among people who have already decided to change their employer and people who think about it often and sometimes. For people who have already decided to change their employer besides the culture of hierarchy, market culture is also in the leading position with the same average value. For people who think sometimes about changing employers besides the culture of hierarchy, the culture of the clan and the culture of the market are also in the leading position with identical average values. The clan culture is dominant among people who really do not think about a new employer or at all.

### 3.3. Normative Commitment

Figure 5 shows the current status of NC in sample enterprises.

#### 3.3.1. Indirect Normative Commitment

The indirect NC was the best assessed by the age group of up to 30 years. The most indirect normatively committed were persons who completed high school. They are followed by persons with Master’s degrees, MBA, Ph.D., or diplomas. Persons with other education have the lowest average value and are consequently the least indirectly normatively committed. According to the three levels of management, the lower management gave the highest score and is followed by middle management. According to the total length of service, the most indirect normatively committed are persons with a total length of service of up to 10 years. Regarding the length of service in the current enterprise, the highest average value was given by employees who have been employed in the current enterprise from 3 to 10 years and more than 20 years. They are followed by employees who have been employed in the current enterprise for up to 3 years. According to the results, employees in micro-enterprises proved to be the most indirect normatively committed. They are followed by employees in large and medium enterprises and lastly by employees in small enterprises. Regarding primary activities of enterprises, there is no difference between average values for indirect NC which is a very interesting and surprising fact.

#### 3.3.2. Direct Normative Commitment

Within the age groups, the direct NC was assessed the worst by those aged 31–40. The most directly normatively committed are persons who completed high school. Employees who are part of the top management gave the highest average value to direct NC; they are followed by employees who are part of middle management. If we consider the total length of service, persons with a total length of service of up to 3 years and over 20 are directly the most committed. The criterion of length of service in the current enterprise is based on the principle that the longer someone is in the current enterprise, the higher its direct NC is, namely: employees who have been in the current enterprise for more than 20 years gave it the highest average value, they are followed by those in the current enterprise from 3 to 20 years, while the worst average values were given by employees who were in the current enterprise for up to 3 years. In micro-enterprises, direct NC came to the fore with the highest average value. It is followed by small enterprises and in the last place medium and large enterprises with the lowest average values. Regarding primary activities of enterprises, there is no difference between average values for direct NC which is a very interesting and surprising fact. Persons who have already decided to change their employer gave the lowest average value to direct NC. The average value of direct NC in all others together (those who think about it or not) is higher.

#### 3.3.3. Common Normative Commitment

In the age groups, the best average values for common NC were given from those under 30 years old. They are followed by persons aged 51 or more. In the case of the highest formal education, it is evident that the most common normatively committed persons are those who completed high school. All three levels of management, as well as professional and other staff, have an identical average value which is a very interesting and surprising fact. Respondents with a total length of service from 10 to 20 years are the least normatively committed. The common NC was most pronounced among the people who have been in the current enterprise the longest, according to our criteria, it is 20 years or more. They are followed by people who have been in current enterprises for 3–10 years. Micro enterprises are at the forefront of the common NC considering the size of enterprises. Regarding manufacturing activity (e.g., manufacturing enterprise), the average value of common NC is the lowest, while in all other activities, common NC is higher. Persons who have already decided to change employers have the lowest average value of a common NC. The average value of the common NC of all others together (those who think about it or not) is higher.

### 3.4. The Impact of Individual Areas and Development of the Model

According to the used statistical methodology and the assumed hypotheses (with sub-hypotheses) in Table 2, Table 3 and Table 4, we present the results of statistically significant impacts. Based on the results, we can confirm all hypotheses as well as all sub-hypotheses. In the area of NC, we marked statements/variables with: Q21a (I think that people these days move from company to company too often), Q21b (I do not believe that a person must always be loyal to his or her organization), Q21c (Jumping from organization to organization does not seem at all unethical to me), Q21d (One of the major reasons I continue to work in this organization is that I believe loyalty is important and therefore feel a sense of moral obligation to remain), Q21e (If I got another offer for a better job elsewhere I would not feel it was right to leave my organization), Q21f (I was taught to believe in the value of remaining loyal to one organization), Q21g (Things were better in the days when people stayed in one organization for most of their careers) and Q21h (I do not think that to be a ‘company man’ or ‘company woman’ is sensible anymore).

It is interesting that OC has an exclusively positive statistically significant impact on SD, while NC has both statistically significant positive and negative impacts on SD. The impact of OC on NC is also both statistically significant positive and negative. The results definitely confirm the extreme importance of both OC and NC for improving SD indicators in supply chains.

### 3.5. Results Application for the New Developed Model

Figure 6 shows the developed model for the development of a sustainable supply chain considering the OC and NC from a logistics perspective. The mentioned model was developed on the basis of various statistical analyses (with an emphasis on confirmed statistically significant positive and negative direct impacts) and confirmed hypotheses H1, H2, and H3 and their sub-hypotheses.

### 3.6. Model Development and Potential Applications to Improve Supply Chain Sustainability

The implementation of the developed model is important in enterprises from the perspectives of OC and NC for the purpose of improving sustainability in their supply chains. The first direction in the use of the model is that it makes the most sense for the management in the enterprises to first identify the OC and NC if they are not known. The method of using the developed model (considering the knowledge or ignorance of the OC and/or NC status) to improve the indicators of SD of the supply chain from a logistics perspective with the help of OC and NC for enterprises or managers in them is as follows:If management in the enterprise is not aware of which type of OC prevails in their enterprise, they should improve the OC in general (all four types of it). The reason is that the OC formed by a factor of four types has a statistically significant positive impact on the SD of supply chains in enterprises (H1), which means that the better OC is the greater positive impact it has on improving SD indicators in supply chains of enterprises.If the dominant type of OC in the enterprise is clan culture, adhocracy culture, market culture, or hierarchy culture, then management should additionally improve them. The reason is that all four types of OC have a statistically significant positive impact on the SD of supply chains in enterprises (H1a, H1b, H1c, and H1d), which means that the better the culture of clan/adhocracy/market/hierarchy is, a greater positive impact it has on improving SD indicators in supply chains of enterprises.If management in the enterprise is not aware of whether their employees are more characterized by direct or indirect NC, then they should focus on the common NC. In the case of a common NC, which is formed by 8 variables (H2), management should be very careful, as it has a statistically positive as well as a negative impact on the SD of supply chains in enterprises. The impact of the two variables (Q21a and Q21e) of NC is positive, which means that the more employees agree with these statements, the better SD is. The impact of variable Q21b is negative, which means that the more employees agree with this statement, the worse SD is. This means that the more employees agree with the variables of NC Q21a and Q21e and the more they disagree with variable Q21b, the greater the positive impact this has on improving the indicators of SD in the supply chains of enterprises.If employees in the enterprise are more characterized by indirect NC (H2a), management should be very careful as this has a statistically positive as well as a negative impact on the SD of supply chains in enterprises. The impact of variable Q21f of indirect NC is positive, which means that the more employees agree with this statement, the better the SD of supply chains in enterprises. The impact of variable Q21b is negative, which means that the more employees agree with this statement, the worse the SD of supply chains in enterprises. This means that the more employees agree with the variable of indirect NC Q21f and the more they disagree with the variable Q21b, the greater the positive impact this has on improving the indicators of SD in the supply chains of enterprises.If employees in the enterprise are more characterized by direct NC (H2b), management should be very careful as this has a statistically positive as well as a negative impact on the SD of supply chains in enterprises. The impact of the three variables (Q21a, Q21d, and Q21e) of direct NC is positive, which means that the more employees agree with these statements, the better the SD of supply chains in enterprises. The impact of variable Q21c is negative, which means that the more employees agree with this statement, the worse the SD of supply chains in enterprises. This means that the more employees agree with the variables of direct NC Q21a, Q21d, and Q21e, and the more they disagree with the variables of Q21c, the greater the positive impact this has on improving the indicators of SD in the supply chains of enterprises.If management in the enterprise is not aware of which type of OC prevails in their enterprise, then they should improve OC in general. The reason is also that the better the OC is, the more employees also agree with the following statements of NC: Q21a, Q21d, Q21e (direct NC), and Q21f (indirect NC). Namely, the OC has a statistically significant positive impact on these four variables (H3). At the same time, we found that these four variables have a statistically significant positive impact on the SD of supply chains. This means that the better the OC is, the better the NC across the four variables, and the greater the positive impact this has on improving the indicators of SD in the supply chains of enterprises.If the dominant type of OC in the enterprise is adhocracy culture, market culture, or hierarchy culture, then management should additionally improve them. The reason is also that the better the mentioned types are, the more employees also agree with the following statements of NC: Q21a, Q21d, Q21e (direct NC), and Q21f (indirect NC). Namely, all three types of OC have a statistically significant positive impact on these four variables (H3b, H3c, and H3d). At the same time, we found that these four variables have a statistically significant positive impact on the SD of supply chains. This means that the better the adhocracy/market/hierarchy culture is, the better the NC across the four variables is, and the greater the positive impact this has on improving the indicators of SD in the supply chains of enterprises.If the dominant type of OC in the enterprise is clan culture, then management should be very careful as the clan culture has a statistically significant positive and negative impact on certain variables of NC, which further affect the SD of supply chains (H3a). Clan culture also has a statistically significant positive impact on the following variables: Q21a, Q21d, Q21e (direct NC), and Q21f (indirect NC). These four variables also have a statistically significant positive impact on the SD of supply chains. This means that the better the clan culture is, the better the NC is across the four variables, and the greater the positive impact this has on improving the indicators of SD in the supply chains of enterprises. Clan culture also has a statistically significant negative impact on variables Q21b (indirect NC) and Q21c (direct NC). These variables also have a statistically significant negative impact on SD. This means that the better the clan culture is, the worse the NC is to both variables, and the greater the positive impact this has on improving the indicators of SD in the supply chains of enterprises.

## 4. Discussion

This paper provided research on the current situation of SD, considering all three dimensions of it, OC and NC in micro, small, medium, and large enterprises. At the same time, the research offers an excellent overview of the situation in micro, small, medium, and large enterprises in relation to individual SDGs. It was found that in the dimensions of SD, the dimension of social development is at the forefront, the highest priority for enterprises is SDG “decent work and economic growth”, the most characteristic type of OC is clan culture, while the level of direct NC is more pronounced than the level of indirect NC. This indicates prioritizing the social dimension, especially in times of huge disruptions, uncertainty, and increasing fear of what is about to happen. Sample enterprises are priority oriented towards SDG “decent work and economic growth” which is extremely good for the whole world because it also emphasizes, e.g., fair payment.

Furthermore, the research proved that OC has a statistically significant direct positive impact on sustainable supply chains, both individually and from the perspective of all types. This means that which type of OC prevails is not so important and that it is necessary to focus on everything (the better the OC and its types, the better the indicators of SD are in the supply chains of enterprises). In any case, it is important to detect the current status of OC and to systematically develop and consolidate it in the future, thus, further increasing sustainability in the supply chains. This is particularly important to implement through enterprise management [84]. The importance of knowing the impact of OC was also emphasized [106], while the positive impact of four types of OC on sustainable performance was also mentioned [85]. OC is related to sustainability [51] and supports its vision [16] which was also definitely confirmed in our study. This means that enterprises in a transition towards sustainable supply chains need to also consider changing their OC to fit best to their strategic goals of achieving more sustainable business which might be opposed to some other measures of achieving sustainability (e.g., employing younger employees that have higher environmental awareness but in general lower NC).

We also found that NC has a statistically significant direct impact on the SD of supply chains. The latter has a direct impact on SD not only positively, like OC, but also negatively. The variables Q21a, Q21d, Q21e, and Q21f have a positive direct impact, while the variables Q21b and Q21c have a negative impact. This means that in order to improve SD in supply chains through NC, it is necessary to develop the highest possible agreement with variables that have a positive impact (higher agreement with these means better SD in supply chains) and the lowest with those that have a negative impact (lower agreement with these means better SD in supply chains). It is certainly the right of employees to have their own opinions and beliefs. In order to increase the sustainable operation of the enterprise and supply chains, it is also important to develop beliefs that have a positive impact on increasing SD or to develop the highest level of commitment in enterprises, which influences and leads in this direction. This can also be developed by, e.g., good relations between all employees (regardless of organizational structure), pleasant working environment, team building, setting clear goals/strategies, considering personal interests of employees, recognizing employee’s efforts, praise (good communication), good pay conditions, supporting proposals from employees for positive changes, constant challenges (e.g., education), or different bonuses. All the above will certainly contribute to greater employee satisfaction and consequently to greater affiliation with existing employers and the long-term future of the enterprise that will undoubtfully be sustainably oriented. This will certainly improve their commitment to the enterprise and the consequent level of NC, which has a positive impact on the SD of supply chains. The link between NC and job satisfaction was also highlighted [107,108,109]. The attitude of management is especially important because it can encourage the NC of employees, which in turn reduces their intention to fluctuate [110] and consequently improves sustainability in the supply chains of enterprises. It was found that an important link exists between sustainability and commitment [68] and that NC plays an important role in logistics and consequently supply chains [15]—both were definitely confirmed in our study.

We also found that OC in the majority part of it has a positive direct impact on NC (individually and considering all four types of it)—only the culture of the clan in the minority part has a negative impact on NC. This means that by developing OC, we can also directly influence the development of the level of NC of employees, which has consequences on the impact on supply chain sustainability. This is certainly further confirmation that the areas of OC and NC in the supply chains of enterprises are extremely important factors in the goal to achieve the improvement of SD indicators. The connection between NC and culture was also pointed out in existing literature [65] and was additionally investigated in our study.

The scientific contribution of the research is especially important for developed enterprises and countries, which identify their prosperity with a sustainable future and whose resources enable or transform such a transformation in order to achieve greater/faster sustainability. The contribution of the study is also important due to the lack of knowledge about the nature of SD, as was found out [111], and because different actions that have a positive impact on today’s world become an inevitable trend in the SD era [112]. Sustainable solutions play a significant role [113] in contributing to global SD [114] and must be supported also by theoretical contribution represented by the newly developed model in this study. Recently, the world economy has faced many unexpected situations, e.g., economic crisis, economic expansion, and special situations such as natural disasters or pandemics (e.g., the COVID-19 pandemic) and new models explaining how to make supply chains more resilient and flexible as well as efficient and sustainable are understudied. In these situations, all stakeholders act as a significant factor, and therefore it is important that the personnel structure in enterprises is stable, and that employees, if necessary, understand the special situation and are willing to cooperate. Therefore, we believe that the scientific contribution of our research will also add value for the segment of supply chain management and of course also have wider impacts. From this perspective, as presented in the developed model, it is even more evident that employees are normatively committed, that this is enabled and supported by the OC in enterprises, and that the entire sustainable supply chain from a logistical perspective works harmoniously, with as little internal disruption as possible. The findings of the research will also enable more efficient and faster adaptation of the supply chains of the future to new challenges related to changes in OC and following new trends and goals of SD.

Short supply chains are becoming more and more popular [115] and in the future a positive change in buying behavior towards short supply chains is expected also due to the COVID-19 pandemic [116]. Therefore, it is necessary to mention the importance of shortening supply chains which are important mainly due to the reduction of business and political risks. Shortening supply chains can also be important due to shorter transport routes (lower fuel consumption), which affects the improvement of indicators of SD, and could also affect different levels of sustainability. Furthermore, we assume that OC is different in, e.g., Asia or Europe and that there is different familiarity and knowledge regarding it. According to our research, depending on the knowledge of the current status of OC, different specific actions can be implemented which enable the most effective improvements of sustainable business all over the world. This is also important because OC was found as one of the elements that has a significant impact on environmental sustainability performance [117,118]. However, this was not part of our research, which focuses mainly on OC and NC in connection with SD which also means that it remains open for further research.

## 5. Conclusions

This paper aimed to provide a model for the development of sustainability in logistics and supply chains to clearly and unequivocally define how it is possible with the impact of OC and NC to improve the level of sustainable supply chains in enterprises. It provides an overview of the current status of SD dimensions, OC, and NC in micro, small, medium, and large enterprises and their link to SDGs by identifying the most common and to prioritize the most important.

The overall findings indicate that further activities to improve sustainability in the supply chains of the micro, small, medium, and large enterprises should also focus on OC and NC. It was found that enterprises can additionally improve sustainability in supply chains by using a developed model to develop a sustainable supply chain from a logistics perspective that is considering the impact of OC and NC. This means that management in enterprises can now systematically carry out activities to promote the types of OC and the statements/variables of NC that have a statistically significant positive impact on SD in their supply chains.

This research provides theoretical and practical implications. The developed model will enrich the theoretical aspect of SD in the fields of study and consequently provide researchers and experts with excellent theoretical starting points for further studies on the supply chain management perspective in the light of green and digital transition, also integrating digitalization as an additional variable. It will also provide excellent starting points because this study is not without limitations, while the concept of sustainability remains one of the world’s greatest challenges. Therefore, it is especially important to investigate the direct impacts of different areas on SD in the supply chain from a logistics or non-logistics perspective, which were not directly considered in this study, e.g., organizational climate, trust, behavior, solidarity, ethics, digitization, or cooperation to extend the treasure trove of knowledge related to improving logistics and supply chains and wider, more general sustainability in the future. The findings in this paper will also provide managers in practice with new insights on how to improve the sustainability of their supply chains. This is especially important because such new models are an essential segment for gaining a long-term competitive advantage on the market with an increasing segment of green customers committed to a sustainable lifestyle and to achieving SDGs.

## Figures and Tables

**Figure 1 ijerph-19-11131-f001:**
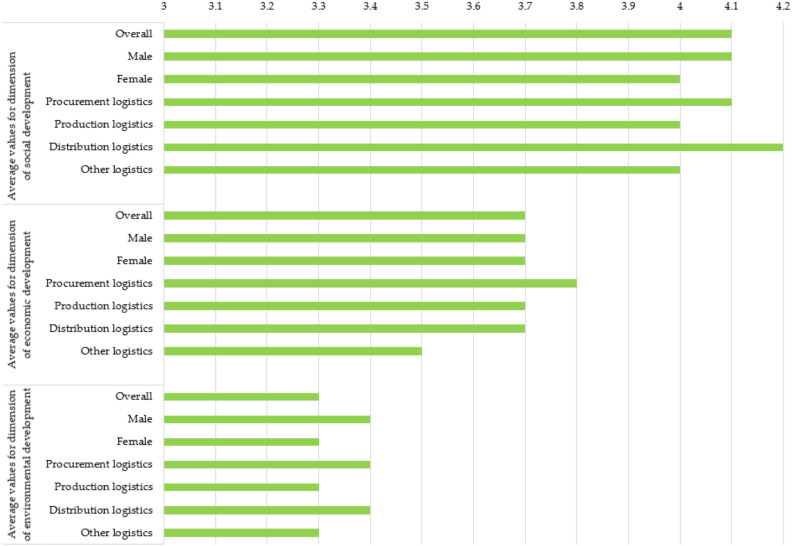
Average values of SD dimensions in sample enterprises (1 means a very low value, 5 means a very high value).

**Figure 2 ijerph-19-11131-f002:**
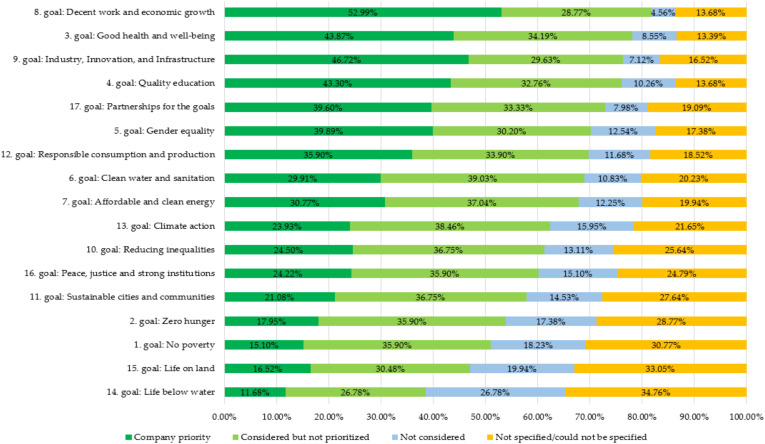
The identified orientation toward specific SDGs in sample enterprises for the overall results.

**Figure 3 ijerph-19-11131-f003:**
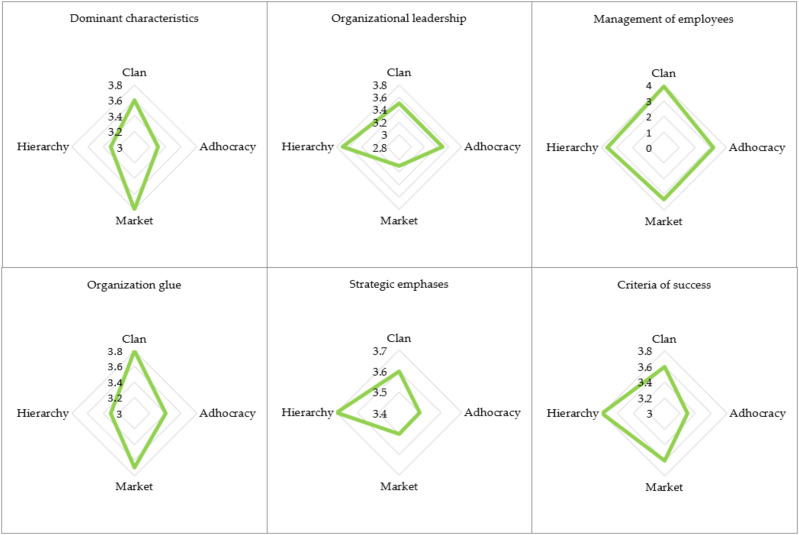
Current status of OC in sample enterprises considering six key dimensions of OC for the overall results.

**Figure 4 ijerph-19-11131-f004:**
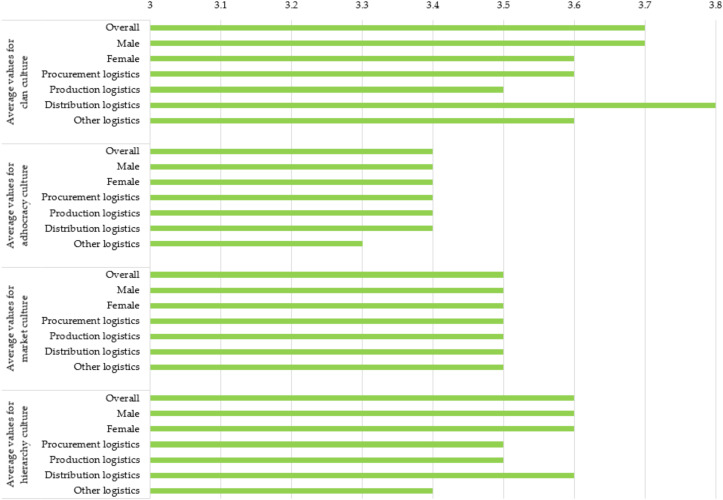
Current status of OC by its types in sample enterprises (1 means a very low value, 5 means a very high value).

**Figure 5 ijerph-19-11131-f005:**
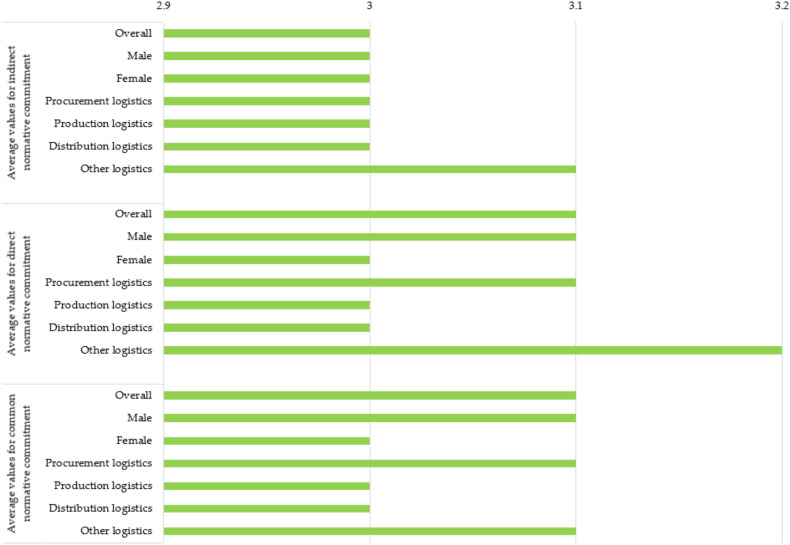
Current status of NC in sample enterprises (1 means a very low value, 5 means a very high value).

**Figure 6 ijerph-19-11131-f006:**
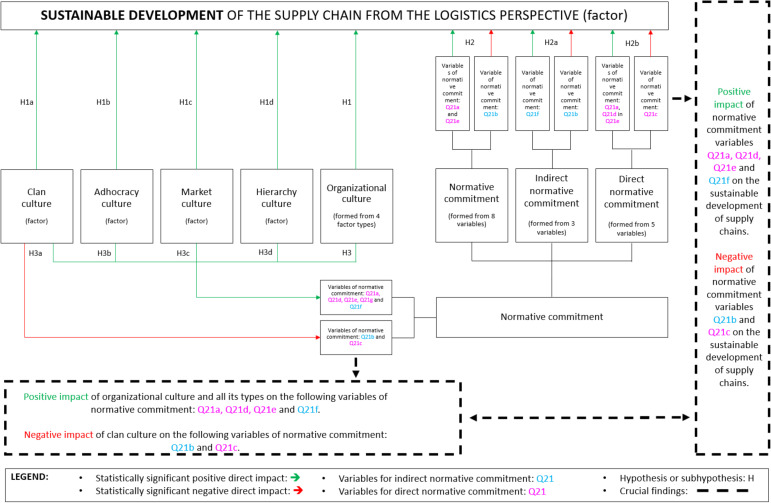
Developed model for improvement of supply chains’ sustainability.

**Table 1 ijerph-19-11131-t001:** Demographic characteristics.

Number	Question	Answer (%)	
1	Gender	Male	62.79%
		Female	37.21%
2	Age	Up to 30 years	17.44%
		From 31 to 40 years	32.95%
		From 41 to 50 years	30.81%
		51 years or older	18.80%
3	Education	High school	25.39%
		Bachelor’s degree/diploma	45.16%
		Master’s degree, MBA, or PhD	27.52%
		Other	1.94%
4	Position in enterprise	Professional staff and others	40.08%
		Lower management	12.94%
		Middle management	28.39%
		Top management	18.58%
5	Type of logistics/logistics subsystems	Procurement logistics	16.08%
		Production logistics	13.99%
		Distribution logistics	52.19%
		Other	17.75%
6	Time of employment	From 0 to 3 years	3.76%
		From 3 to 10 years	27.14%
		From 10 to 20 years	32.78%
		20 or more years	36.33%
7	Time of employment in the current enterprise	From 0 to 3 years	21.50%
		From 3 to 10 years	34.45%
		From 10 to 20 years	26.72%
		20 or more years	17.33%
8	Enterprise size	Micro enterprise	14.26%
		Small enterprise	21.70%
		Medium enterprise	28.09%
		Large enterprise	35.96%
9	Main activity of enterprise	Manufacturing activity	35.74%
		Transport and storage activity	42.98%
		Other business activities	21.28%
10	Considering changing your job/employment	Yes, already decided	3.62%
		Yes, often	5.74%
		Yes, sometimes	26.38%
		Not really	39.57%
		Not at all	24.68%
11	Enterprise location	Inside the European Union—21 countries (Austria, Bulgaria, Croatia, Cyprus, Czech Republic, Denmark, Finland, France, Germany, Hungary, Ireland, Italy, Latvia, Lithuania, the Netherlands, Poland, Romania, Slovakia, Slovenia, Spain, and Sweden)	86.81%
		Outside the European Union—13 countries (Albania, Bosnia and Herzegovina, China, Egypt, Mexico, North Macedonia, Philippines, Russia, Serbia, South Africa, Turkey, Ukraine, and the United States of America)	11.70%
		Other (answers, e.g., global, worldwide, multinational, etc.)	1.49%

**Table 2 ijerph-19-11131-t002:** Results of statistical analysis for hypothesis H1 and its sub-hypotheses.

Hypothesis and Sub-Hypotheses	Used for Statistical Analysis	Standardized Coefficient (β)	t	*p*	Impacts	
					Statistically Significant?	Positive?
H1	Factor OC on factor SD	0.670	17.142	0.000	Yes	Yes
H1a	Factor clan culture on factor SD	0.635	15.585	0.000	Yes	Yes
H1b	Factor adhocracy culture on factor SD	0.584	13.649	0.000	Yes	Yes
H1c	Factor market culture on factor SD	0.390	8.031	0.000	Yes	Yes
H1d	Factor hierarchy culture on factor SD	0.547	12.394	0.000	Yes	Yes

**Table 3 ijerph-19-11131-t003:** Results of statistical analysis for hypothesis H2 and its sub-hypotheses.

Hypothesis and Sub-Hypotheses	Used for Statistical Analysis	Standardized Coefficient (β)	t	*p*	Impacts	
					Statistically Significant?	Positive?
H2	Variable Q21a of NC on factor SD	0.168	3.146	0.002	Yes	Yes
	Variable Q21b of NC on factor SD	−0.121	−2.173	0.030	Yes	No
	Variable Q21c of NC on factor SD	−0.052	−0.966	0.335	No	No
	Variable Q21d of NC on factor SD	0.100	1.704	0.089	No	Yes
	Variable Q21e of NC on factor SD	0.148	2.703	0.007	Yes	Yes
	Variable Q21f of NC on factor SD	0.105	1.713	0.088	No	Yes
	Variable Q21g of NC on factor SD	−0.060	−1.044	0.297	No	No
	Variable Q21h of NC on factor SD	−0.052	−0.992	0.322	No	No
H2a	Variable Q21b of NC on factor SD	−0.148	−2.789	0.006	Yes	No
	Variable Q21f of NC on factor SD	0.220	4.257	0.000	Yes	Yes
	Variable Q21h of NC on factor SD	−0.053	−1.008	0.314	No	No
H2b	Variable Q21a of NC on factor SD	0.175	3.253	0.001	Yes	Yes
	Variable Q21c of NC on factor SD	−0.118	−2.372	0.018	Yes	No
	Variable Q21d of NC on factor SD	0.151	2.711	0.007	Yes	Yes
	Variable Q21e of NC on factor SD	0.155	2.866	0.004	Yes	Yes
	Variable Q21g of NC on factor SD	−0.034	−0.618	0.537	No	No

**Table 4 ijerph-19-11131-t004:** Results of statistical analysis for hypothesis H3 and its sub-hypotheses.

Hypothesis and Sub-Hypotheses	Used for Statistical Analysis	Standardized Coefficient (β)	t	*p*	Impacts	
					Statistically Significant?	Positive?
H3	Factor OC on variable Q21a of NC	0.260	5.168	0.000	Yes	Yes
	Factor OC on variable Q21b of NC	−0.100	−1.929	0.055	No	No
	Factor OC on variable Q21c of NC	−0.040	−0.761	0.447	No	No
	Factor OC on variable Q21d of NC	0.303	6.087	0.000	Yes	Yes
	Factor OC on variable Q21e of NC	0.287	5.747	0.000	Yes	Yes
	Factor OC on variable Q21f of NC	0.278	5.541	0.000	Yes	Yes
	Factor OC on variable Q21g of NC	0.161	3.117	0.002	Yes	Yes
	Factor OC on variable Q21h of NC	−0.023	−0.442	0.659	No	No
H3a	Factor clan culture on variable Q21a of NC	0.228	4.493	0.000	Yes	Yes
	Factor clan culture on variable Q21b of NC	−0.165	−3.203	0.001	Yes	No
	Factor clan culture on variable Q21c of NC	−0.130	−2.515	0.012	Yes	No
	Factor clan culture on variable Q21d of NC	0.280	5.597	0.000	Yes	Yes
	Factor clan culture on variable Q21e of NC	0.239	4.723	0.000	Yes	Yes
	Factor clan culture on variable Q21f of NC	0.257	5.105	0.000	Yes	Yes
	Factor clan culture on variable Q21g of NC	0.110	2.117	0.035	Yes	Yes
	Factor clan culture on variable Q21h of NC	−0.099	−1.897	0.059	No	No
H3b	Factor adhocracy culture on variable Q21a of NC	0.268	5.328	0.000	Yes	Yes
	Factor adhocracy culture on variable Q21b of NC	−0.085	−1.636	0.103	No	No
	Factor adhocracy culture on variable Q21c of NC	−0.004	−0.077	0.938	No	No
	Factor adhocracy culture on variable Q21d of NC	0.312	6.289	0.000	Yes	Yes
	Factor adhocracy culture on variable Q21e of NC	0.313	6.308	0.000	Yes	Yes
	Factor adhocracy culture on variable Q21f of NC	0.298	5.989	0.000	Yes	Yes
	Factor adhocracy culture on variable Q21g of NC	0.196	3.833	0.000	Yes	Yes
	Factor adhocracy culture on variable Q21h of NC	0.015	0.281	0.779	No	Yes
H3c	Factor market culture on variable Q21a of NC	0.131	2.541	0.011	Yes	Yes
	Factor market culture on variable Q21b of NC	0.024	0.453	0.651	No	Yes
	Factor market culture on variable Q21c of NC	0.053	1.010	0.313	No	Yes
	Factor market culture on variable Q21d of NC	0.157	3.052	0.002	Yes	Yes
	Factor market culture on variable Q21e of NC	0.188	3.658	0.000	Yes	Yes
	Factor market culture on variable Q21f of NC	0.177	3.447	0.001	Yes	Yes
	Factor market culture on variable Q21g of NC	0.105	2.019	0.044	Yes	Yes
	Factor market culture on variable Q21h of NC	0.003	0.049	0.961	No	Yes
H3d	Factor hierarchy culture on variable Q21a of NC	0.208	4.081	0.000	Yes	Yes
	Factor hierarchy culture on variable Q21b of NC	−0.088	−1.701	0.090	No	No
	Factor hierarchy culture on variable Q21c of NC	−0.040	−0.769	0.442	No	No
	Factor hierarchy culture on variable Q21d of NC	0.222	4.362	0.000	Yes	Yes
	Factor hierarchy culture on variable Q21e of NC	0.185	3.597	0.000	Yes	Yes
	Factor hierarchy culture on variable Q21f of NC	0.160	3.098	0.002	Yes	Yes
	Factor hierarchy culture on variable Q21g of NC	0.105	2.020	0.044	Yes	Yes
	Factor hierarchy culture on variable Q21h of NC	0.008	0.161	0.872	No	Yes

## Data Availability

Some or all data and models that support the findings of this study are available from the corresponding author upon reasonable request. The data are not publicly available due to potential further publications.

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
