# Peer review of "Boosting Sustainable Operations with Sustainable Supply Chain Modeling: A Case of Organizational Culture and Normative Commitment"

_ijerph, 2022, doi:10.3390/ijerph191711131_

Round 1

Reviewer 1 Report

The article deals with the important topic of Boosting Sustainable Operations with Sustainable Supply Chain Modeling. The Case of Organizational Culture and Normative Commitment was selected as an example.

The importance of sustainable development is emphasized in many studies and is currently a development paradigm in many countries of the world. Therefore, this is a current issue and therefore I assess it positively.

In the manuscript in the introduction, the aim of the research was correctly constructed, which was then implemented in the following chapters. There were also three main hypotheses and a sub-hypothesis in each of them. The question remains whether such a large number is necessary?

Supply chains are referenced in many places in the manuscript, which is of course justified in the area of ​​logistics. However, there was no information on how shortening supply chains could have an impact on sustainable development. This is particularly important in view of the disturbance that exists and the experiences that the Covid-19 pandemic has brought to many companies.

The analyzes are rather correct. However, I am wondering how the results are presented in chapter 3.6. Most of the content indicated there is conditional sentences beginning with "if ...".

The conclusions contain a correct list of the most important issues resulting from the conducted analyzes. I also positively assess the fact that the limitations of the conducted research have been indicated and the proposals for future analyzes.

In the discussion chapter, there are very few references to the results of other studies. There are only a few of them throughout the chapter. It should be supplemented, because it is in this chapter that the results should be scientifically discussed with others known from the literature on the subject.

The results of own research conducted in a group of enterprises can be compared, for example, with the results of the way sustainable development is understood by other social groups. Interesting comparisons on this subject can be found, for example, in The Idea of ​​Sustainable Development and the Possibilities of Its Interpretation and Implementation. Energies 2022, 15, 5394 or many others.

Figure 1 lacks the range of the rating scale (1-5?) And lacks information, as evidenced by lower and higher values.

Figures 1, 2, 3, 6 go beyond the margins. Correct their formatting. The same problem applies to tables 2, 3, 4.

There are minor errors in the way the publication is cited. On p. 2 it is eg [19-20], but it should be [19, 20] correctly.

The list of bibliographic items was prepared contrary to the guidelines for authors. This needs to be corrected.

Author Response

Reply to reviewer #1:

Thank you for reviewing our manuscript and presenting your perspective and proposals for potential improvements of it. The English text has already been proofread, however, due to significant changes and improvements, it was again additionally checked. We are sending answers to your comments by points:

  1. The article deals with the important topic of Boosting Sustainable Operations with Sustainable Supply Chain Modeling. The Case of Organizational Culture and Normative Commitment was selected as an example.

Answer: Thank you for marking our topic as important. It is correct - Organizational Culture and Normative commitment are selected as an example.

  1. The importance of sustainable development is emphasized in many studies and is currently a development paradigm in many countries of the world. Therefore, this is a current issue and therefore I assess it positively.

Answer: Thank you for the positive assessment.

  1. In the manuscript in the introduction, the aim of the research was correctly constructed, which was then implemented in the following chapters. There were also three main hypotheses and a sub-hypothesis in each of them. The question remains whether such a large number is necessary?

Answer: We completely agree with you that maybe such a large number of hypotheses and a sub-hypothesis are not necessarily. Nevertheless, we decided on a larger number of hypotheses and sub-hypotheses because we wanted to clearly show that we studied different impacts and of course which ones. A larger number of hypotheses and sub-hypotheses were also used because we wanted to study the field of research on the one hand systematically and on the other hand as comprehensively as possible. We believe that each hypothesis and/or sub-hypothesis brings specific added value to the research results.

  1. Supply chains are referenced in many places in the manuscript, which is of course justified in the area of ​​logistics. However, there was no information on how shortening supply chains could have an impact on sustainable development. This is particularly important in view of the disturbance that exists and the experiences that the Covid-19 pandemic has brought to many companies.

Answer: Exactly, information about shortening supply chains was missing. We also agree that it could have an impact on sustainable development and because of that, we added a few words regarding it in the discussion chapter.

  1. The analyzes are rather correct. However, I am wondering how the results are presented in chapter 3.6. Most of the content indicated there is conditional sentences beginning with "if ...".

Answer: In the mentioned chapter, we used "IF" because the applicability of the results is presented for different knowledge and familiarity of the areas of organizational culture and normative commitment in practice. Depending on the knowledge and familiarity, enterprises or managers can then focus on how to improve the indicators of sustainable development in their supply chains with the help of the research results.

  1. The conclusions contain a correct list of the most important issues resulting from the conducted analyzes. I also positively assess the fact that the limitations of the conducted research have been indicated and the proposals for future analyzes.

Answer: Thank you for your comment.

  1. In the discussion chapter, there are very few references to the results of other studies. There are only a few of them throughout the chapter. It should be supplemented, because it is in this chapter that the results should be scientifically discussed with others known from the literature on the subject.

Answer: In accordance with your recommendation, we again reviewed the discussion chapter in detail and upgraded it appropriately.

  1. The results of own research conducted in a group of enterprises can be compared, for example, with the results of the way sustainable development is understood by other social groups. Interesting comparisons on this subject can be found, for example, in The Idea of ​​Sustainable Development and the Possibilities of Its Interpretation and Implementation. Energies 2022, 15, 5394 or many others.

Answer: We reviewed the article published in Energies with the title ‘’The Idea of Sustainable Development and the Possibilities of Its Interpretation and Implementation’’ and we agree that it fits very well in our article. We decided to use the findings of your suggested article for our discussion chapter. In addition, we also included some other sources in the discussion section, as you suggested us.

  1. Figure 1 lacks the range of the rating scale (1-5?) And lacks information, as evidenced by lower and higher values.

Answer: In some pictures, we didn’t have this information because we specifically emphasized it in chapter 2.1 in which we wrote: ''The responses were recorded mainly by the five-point Likert scale. Consent rates ranged from one, which meant disagreement at all, to a maximum of five, which meant very strong agreement''. Regardless of this information in chapter 2.1, we have considered your recommendation and added additional information to Figure 1 and also to Figure 4 and Figure 5. In all three mentioned Figures now we wrote short note ''1 means very low value, 5 means very high value'' to further improve the clarity of individual Figures.

  1. Figures 1, 2, 3, 6 go beyond the margins. Correct their formatting. The same problem applies to tables 2, 3, 4.

Answer: Some of the Figures and Tables are beyond the margins only for the review process because we wanted to enable editors and reviewers to better data review. If the article will be accepted for publication in the International Journal of Environmental Research and Public Health then all Figures and all Tables will be uploaded for publication separately in the highest possible quality.

  1. There are minor errors in the way the publication is cited. On p. 2 it is eg [19-20], but it should be [19, 20] correctly.

Answer: In accordance with your instructions, we have corrected such wrong citations for 19 and 20 and also for all other misquotes like that, e.g. 13, 14, 32, 33, etc.

  1. The list of bibliographic items was prepared contrary to the guidelines for authors. This needs to be corrected.

Answer: We corrected the list of all bibliographic items which are now in accordance with the guidelines for authors.

Reply to reviewer #3:

Answer: The authors would like to thank you for reviewing our manuscript and especially for a very positive review and proposed acceptance of the paper even in the primary form.

Reviewer 2 Report

This paper is of sound quality on a subject deserving the Journal's attention. This study attempts to how different types of organizational culture and normative commitment impact sustainability and each other in business logistics and supply chains. Developing a conceptual model to manage this challenge, this paper identified valuable especially for managers to get better information on how to improve sustainability not just by integrating green technologies but mainly by changing culture, attitude, and perception in their enterprises as a new insights. Overall, the paper is well written and well structured, therefore it is easy to follow and builds a clear conclusion from the data. Generally well written but requires some editing and revision.

In literature review, this study well reviewed prior research on SD and sustable operations in logistics. However, logistics is a part of supply chain. they can not represent supply chain. The author requires additional justification for sustainable operations in supply chain.

Research design, data collection process and data analysis method are appropriate. Justification of data collection periods would be explained, additionally.

Line 83 requires sentence check.

to improve readability, I would recommend inserting research model of conceptual model embracing all of hypothesis.

In Table 3, 4, 5, the name of each valuable should be presented, rather than a,b,c,d...types.

The processes for data analysis are appropriate and the results of it are clearly described. However, this paper just described the results of data analysis. To improve the quality of this study, author(s) need to extract more clear implications in both theoretical and practical perspectives as a discussion of the results. Additional explanations are required to link the results of data analysis and conclusions. Research conclusion (practical implication) part is weak, focusing on data analysis (enumerate bits of information). Additional explanations incorporating theoretical and practical are required.

The quality of communication is appropriate. Generally, well written but requires some editing and revision.

Author Response

Reply to reviewer #2:

Thank you for reviewing our manuscript and presenting your perspective and proposals for potential improvements of it. The English text has already been proofread; however, due to significant changes and improvements, it was again additionally checked. We are sending answers to your comments by points:

  1. This paper is of sound quality on a subject deserving the Journal's attention. This study attempts to how different types of organizational culture and normative commitment impact sustainability and each other in business logistics and supply chains. Developing a conceptual model to manage this challenge, this paper identified valuable especially for managers to get better information on how to improve sustainability not just by integrating green technologies but mainly by changing culture, attitude, and perception in their enterprises as a new insight. Overall, the paper is well written and well structured, therefore it is easy to follow and builds a clear conclusion from the data. Generally well written but requires some editing and revision.

Answer: Thank you for marking our article as a subject deserving the Journal's attention.

  1. In the literature review, this study well reviewed prior research on SD and sustainable operations in logistics. However, logistics is a part of the supply chain. they can not represent the supply chain. The author requires additional justification for sustainable operations in supply chain.

Answer: It is absolutely true that supply chains are a combination of different combinations of different logistics operations in enterprises. Our literature review is written in this area with a focus on the supply chain, whereas the logistics of the aforementioned are beside the supply chains and is not the central focus. The focus is more on (sustainable) supply chains. Based on your comment, we decided to correct the initial part of the literature review a little.

  1. Research design, data collection process and data analysis method are appropriate. Justification of data collection periods would be explained additionally.

Answer: According to your recommendation, we updated data collection periods.

  1. Line 83 requires sentence check.

Answer: According to your initiative, we made improvements to some of the sentences in chapters 1.2 and 1.3.

  1. To improve readability, I would recommend inserting a research model of conceptual model embracing all of hypotheses.

Answer: The conceptual model which embraces all of the hypotheses is shown in Figure 6 in chapter 3.5. However, it is true that the mentioned conceptual model also shows some other matters, e.g. statistically significant positive direct impacts, crucial findings, etc. If you believe that inserting an additional conceptual model is important, then we can add it, but we think there is no need. Mainly because the article is already very long, and we would not like to stretch it even further.

  1. In Table 3, 4, 5, the name of each valuable should be presented, rather than a, b, c,d...types.

Answer: In Tables 3, 4 and 5 we use ''a, b, c, d... '' on purpose. If we would not use ''a, b, c, d... '', the tables, which are very extensive, would be more difficult to read (reason: large tables size). The scope of the article, which is already quite large, would also increase further. Hypotheses, sub-hypotheses and labels ''a, b, c, d...'' are precisely defined in the methodology and above Table 2.

  1. The processes for data analysis are appropriate and the results of it are clearly described. However, this paper just described the results of data analysis. To improve the quality of this study, author(s) need to extract more clear implications in both theoretical and practical perspectives as a discussion of the results. Additional explanations are required to link the results of data analysis and conclusions. Research conclusion (practical implication) part is weak, focusing on data analysis (enumerate bits of information). Additional explanations incorporating theoretical and practical are required.

Answer: Thanks for the recommendation. According to your recommendation and recommendations from other reviewers, we extend and additionally upgraded the discussion chapter.

  1. The quality of communication is appropriate. Generally, well written but requires some editing and revision.

Answer: Thank you for your recommendation. Based on the recommendations obtained as part of the review, the article was additionally corrected and improved (upgraded).

Reply to reviewer #3:

Answer: The authors would like to thank you for reviewing our manuscript and especially for a very positive review and proposed acceptance of the paper even in the primary form.

Reviewer 3 Report

After getting familiar with the reading-matter of the reviewed article I state the following:

1. The article examines how different types of organizational culture (OC) and normative commitment (NC) affect sustainable development (SD) and each other in business logistics and supply chains, and developed a conceptual model to manage this challenge.
2. The presented issue is very important as the society must be aware that the concept of sustainable development remains one of the greatest challenges in the world.
3. The analysis of the obtained results is carried out very well.
4. The article would be more valuable if the Authors also included numerical values, and not only percentages, which - after all - slightly distort the real picture.

Author Response

Reply to reviewer #3:

Answer: The authors would like to thank you for reviewing our manuscript and especially for a very positive review and proposed acceptance of the paper even in the primary form.